# Can the Double Exchange Cause Antiferromagnetic Spin Alignment?

**Andrew Palii** [1,*], **Juan M. Clemente-Juan** [2,*] , **Sergey Aldoshin** [1] , **Denis Korchagin** [1] ,
**Evgenii Golosov** [1], **Shmuel Zilberg** [3] **and Boris Tsukerblat** [3,*]

[1]  Laboratory of Molecular Magnetic Nanomaterials, Institute of Problems of Chemical Physics,
    Academician Semenov Avenue 1, 142432 Chernogolovka, Moscow Region, Russia; sma@icp.ac.ru (S.A.);
    korden@icp.ac.ru (D.K.); golosov@icp.ac.ru (E.G.)
[2]  Instituto de Ciencia Molecular, Universidad de Valencia, 46980 Paterna, Spain
[3]  Department of Chemical Sciences, Ariel University, Ariel 40700, Israel; shmuel.zilberg@gmail.com
[*]  Correspondence: andrew.palii@uv.es (A.P.); juan.m.clemente@uv.es (J.M.C.-J.); tsuker@bgu.ac.il (B.T.)

**Abstract:** The effect of the double exchange in a square-planar mixed-valence $d^{n+1} - d^{n+1} - d^n - d^n$–type tetramers comprising two excess electrons delocalized over four spin cores is discussed. The detailed analysis of a relatively simple $d^2 - d^2 - d^1 - d^1$–type tetramer shows that in system with the delocalized electronic pair the double exchange is able to produce antiferromagnetic spin alignment. This is drastically different from the customary ferromagnetic effect of the double exchange which is well established for mixed-valence dimers and tetramers with one excess electron or hole. That is why the question "Can double exchange cause antiferromagnetic spin alignment?" became the title of this article. As an answer to this question the qualitative and quantitative study revealed that due to antiparallel directions of spins of the two mobile electrons which give competitive contributions to the overall polarization of spin cores, the system entirely becomes antiferromagnetic. It has been also shown that depending on the relative strength of the second-order double exchange and Heisenberg–Dirac–Van Vleck exchange the system has either the ground localized spin-triplet or the ground delocalized spin-singlet.

**Keywords:** mixed-valence; electron transfer; double exchange; magnetic exchange; tetrameric mixed valence clusters; quantum cellular automata

## 1. Introduction

Mixed-valence (MV) compounds have been discovered more than a century ago and proved to be in focus of a wide range of chemistry and physics forming basis for the concept of intramolecular electron transfer. Fundamental contributions formulated as semiclassical vibronic Marcus–Hush theory and quantum mechanical treatments of the electron transfer by Piepho–Krauzs–Schatz (PKS theory) laid the foundations for understanding of chemical transformations and spectroscopic properties of molecules and crystals. A detailed survey of the development and numerous applications of the classical theoretical concepts of electron transfer in chemistry and spectroscopy is given in Ref. [1].

A milestone in the development of the concept of mixed valence was the discovery of the double exchange [2–4] as an origin of the ferromagnetic ordering in perovskite structure $(La_XCa_{1-x})(Mn^{III}Mn^{IV}_{1-x})O_3$ containing MV fragment $Mn^{III}$-$O^{2-}$-$Mn^{IV}$. The double exchange can be referred to as a spin-dependent electron transfer over the magnetic metal sites whose spins are polarized by the mobile electron giving rise to the ferromagnetic spin alignment. Migration of these ideas from solid state physics to chemistry was stimulated by the study of molecular systems of biological significance, such as two-iron ($Fe^{2+}$-$Fe^{3+}$) ferredoxin, protein with $[Fe_3S_4]$ core [5] and also of other

complex polynuclear MV systems like reduced polyoxometalates with Keggin and Wells–Dawson structures [6–8]. Molecular applications of the concept of double exchange gave impact to the generalization of the theory as applied to the systems with arbitrary number of mobile electrons and to large multicenter systems [9,10]. Success of the theory was proved by the treatment of polynuclear MV clusters such as hexanuclear octahedral clusters $[Fe_6(\mu_3\text{-}X)_8(PEt_3)_6]^+$ (X = S, Se and Et = $C_2H_5$) [11] and giant reduced polyoxovanadates $[V_{18}O_{48}]^{n-}$ ($n = 4 \div 18$) [12,13]. Finally, formulation of the symmetry assisted approach to the solution of multidimensional vibronic problem in nanoscopic MV systems completed the theoretical development of this stage of the field [14,15].

In recent years, the well outlined field of magnetic MV molecules has received a strong impulse for new development, caused by the emerging problem of the so-called quantum cellular automata (QCA). The QCA technology assumes that the binary information is stored in charge distributions (rather than in quantum states of qubits employed in quantum computing schemes) in the QCA cells and is transmitted via Coulomb forces. The initial proposal in the area of QCA has been based on the use of quantum dots to compose cells coupled via Coulomb interaction to form a cellular automata architecture [16–21]. Each such cell typically consists of four quantum dots situated in the vertices of a square and two excess electrons tunneling between the dots. The idea of molecular QCA within which the cells are represented by molecules rather than by the arrays of quantum dots is expected to result in further miniaturization of QCA devices and also gives a number of important advantages [19–21] (see review [22]).

One can expect that MV complexes containing mobile excess electrons can be viewed as natural candidates for molecular cells. In particular, tetrameric MV system with two mobile electrons can be bistable and hence be able to encode binary information 0 and 1 in the two charge distributions. To ensure a proper action of the QCA cell, it should be sensitive to the external control, which means that switching between the two charge configurations should occur in an abrupt nonlinear manner. The problem of the rational design of the cells based on MV molecules is of crucial importance for the area of molecular QCA [23–28]. All molecules proposed up to now as candidate of cells, belong to MV systems in which the mobile charges are delocalized over the network of diamagnetic sites. In this context the idea of using MV systems involving magnetic ions seems to be tempting due to potentiality to employ not only charges as carriers of information but also spin degrees of freedom. Anticipating the study of the functional properties of the molecular cell, in the present article we examine square planar transition metal MV tetramers of $d^{n+1} - d^{n+1} - d^n - d^n$ type in which both double exchange and Heisenberg–Dirac–Van Vleck (HDVV) exchange are operative. The detailed results are given for the model system $d^2 - d^2 - d^1 - d^1$ in which two electrons are shared among four spin-1/2 cores.

## 2. Basic Model for a Mixed-Valence Tetrameric Square-Planar System

We consider a square-planar MV unit composed of four high-spin paramagnetic centers, which can be referred to as "spin cores" and two excess electrons, shared among these centers. This cluster belongs to the $D_{4h}$ point group. There are six possible distributions of the two excess electrons over the four sites as shown in Figure 1. In two of these distributions (electronic configurations) the extra electrons are localized on the antipodal sites forming diagonals of the square, while in the remaining four configurations the electrons occupy neighboring positions forming its edges. The two diagonal electronic configurations (denoted as $D_1$ and $D_2$ in Figure 1) minimize the Coulomb energy of the electronic repulsion due to the fact that in these configurations the two electrons occupy the most distant positions from each other and thus form the ground Coulomb manifold, while the remaining four configurations ($D_3 \ldots D_6$) with shorter interelectronic distance give rise to the excited Coulomb levels separated from the ground manifold by the energy gap $U$. Note that the two ground diagonal configurations are transformed to each other under the action of the operation of rotation around $C_4$ axis within the $D_{4h}$ point group, as well as the four excited configurations. At the same time, the excited configurations cannot be obtained from the ground ones by the $D_{4h}$ group operations thus showing that these two kinds of configurations are physically different.

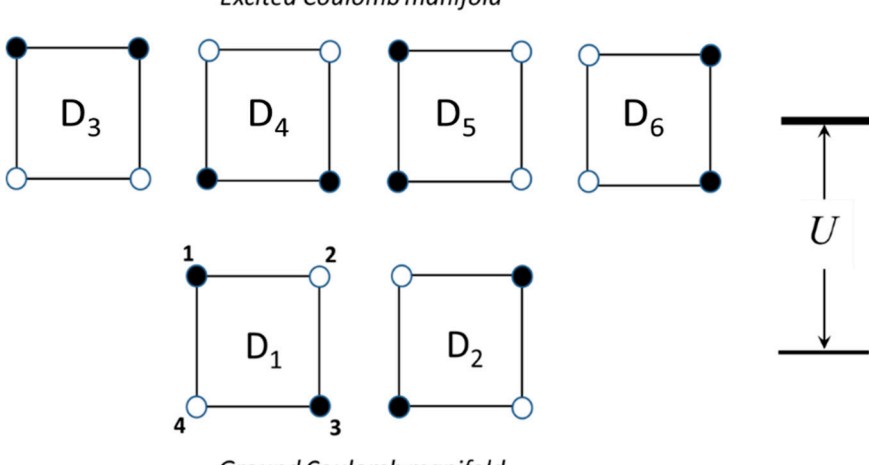

**Figure 1.** Numeration of the sites of the square-planar tetrameric cluster with two mobile electrons, two antipodal distributions denoted as $D_1$ and $D_2$ and four excited neighboring distributions $D_3 \ldots D_6$. The sites occupied by the extra electrons are shown by black balls, the spin cores are indicated as white balls.

The overall double exchange is determined by the one-electron transfer processes occurring from $d^{n+1}$ to $d^n$ centers. As distinguished from the previous considerations of a bi-electronic square planar molecules proposed as cells for QCA applications, [29,30] in the system under consideration the excess electrons jump over the spin cores, and each such electron hopping results in the change of the energy of the tetramer by the Coulomb energy $U$. As distinguished from the previous consideration [29,30] of a bi-electronic square planar tetramer, in the system under consideration the extra electrons jump over spin sites which just constitutes the basis for considering the double exchange. For the sake of definiteness, we will focus on the case of $n \leq 4$ (less than half-filled d-shells). We assume that the $d^n$-ion is the high-spin one and its spin is $S_0 = n/2$. When the excess electron is trapped in some metal site (i.e., this site is occupied by the $d^{n+1}$-ion) its spin is coupled ferromagnetically with the core spin $S_0$ to give the spin $S_0 + 1/2$ as schematized in Figure 2a,b. For the sake of simplicity only the transfer processes between the neighboring sites located along the sides of the square tetramer are assumed to be nonvanishing, consequently $t$ is the transfer parameter. Figure 2a,b shows the transfer processes which produce mixing of the two kinds of charge configurations, for example, mixing of the ground neighboring $D_3$ ($d_1^{n+1} - d_2^{n+1} - d_3^n - d_4^n$) configuration with the antipodal excited $D_1$ ($d_1^{n+1} - d_2^n - d_3^{n+1} - d_4^n$) one. In Figure 2c the orbital scheme illustrating the transfer of the excess electron from the site $i$ to the site $k$ is shown for the simplest case when $n = 1$. Electrons of spin cores occupy orbitals denoted as $\varphi_i$ and $\varphi_k$, while the excess electron may move over the upper orbitals denoted as $\psi_i$ and $\psi_k$ resulting in the polarization of the spin cores in accordance with the conventional double exchange mechanism. In general case each metal site forming the MV tetramer should contain $n + 1$ orbitals, with $n$ of these orbitals (core orbitals) being single occupied and the highest $(n + 1)$-th orbital being either empty or single occupied depending on the position of the excess electron. Note that in the present consideration of the double exchange the excited non-Hund states of each ion are assumed to be separated from the ground Hund states by the energy gaps strongly exceeding both the value of the electron transfer parameters and the Coulomb energy $U$. Under such assumption (that seems to be reasonable in many cases) one can truncate the double exchange problem defining it within the space involving only the ground states of each ions as is it is usually accepted in the modelling the properties of MV clusters exhibiting double exchange.

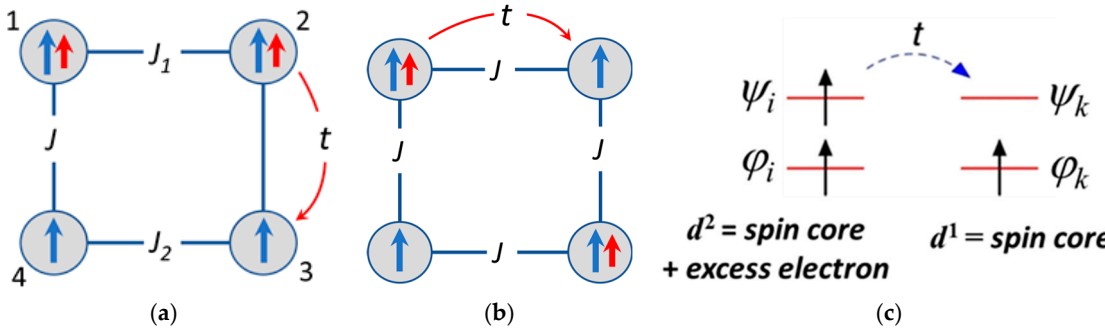

**Figure 2.** Schemes of the magnetic sites, exchange and transfer parameters for the tetrameric MV cluster with two mobile electrons shown for the neighboring Coulomb configuration $D_3$ (**a**) and for distant configuration $D_1$ (**b**), and also the orbital scheme of the one electron transfer between the two sites $i$ and $k$ shown for the simplest case of one-electron spin cores. $\phi$ are the orbitals occupied by the localized electrons (spin-core orbitals), and $\psi$ are the upper orbitals available for the transfer of the excess electron (**c**).

The Hamiltonian describing the double exchange and the intercenter Coulomb repulsion between the two excess electrons in the square planar MV cluster is the following:

$$\hat{H} = t \sum_{i<k,\sigma} (1 - \delta_{k,i+2})\left(\hat{c}^+_{\psi_i \sigma}\hat{c}_{\psi_k \sigma} + \hat{c}^+_{\psi_k \sigma}\hat{c}_{\psi_i \sigma}\right) + \sum_{i<k} U_{ik}\hat{n}_{\psi_i}\hat{n}_{\psi_k} \equiv \hat{H}_{DE} + \hat{H}_0 \tag{1}$$

where the first term $\hat{H}_{DE}$ describes the one-electron transfer between the nearest neighboring sites $i$ and $k$, interrelated with the double exchange, while $\hat{H}_0$ represents the Coulomb repulsion term. In Equation (1) $\hat{c}^+_{\psi_i \sigma}$ and $\hat{c}_{\psi_k \sigma}$ are the creation and annihilation operators, $\sigma$ is the spin projection, $\hat{n}_{\psi_i}$ are the extra electron occupation number operators. On the first glance the Hamiltonian, Equation (1), looks quite similar to the Hubbard Hamiltonian, which also includes the transfer term and the Coulomb repulsion term. However, these two Hamiltonians are quite different in their nature. Indeed, the Hubbard Hamiltonian acts within the space comprising configurations with two excess electrons per metal site and the parameters $U_{ii}$ involved in such Hamiltonian represent the on-site interelectronic Coulomb energies, while in the Hamiltonian, Equation (1), the electronic configurations with two excess electrons occupying the same metal site are excluded, and the parameters $U_{ik}$ describe the Coulomb repulsion between the excess electrons occupying different metal sites $i$ and $k$. According to Figure 1 the Hamiltonian, Equation (1), contains two different Coulomb energies $U_d \equiv U_{13} = U_{24}$ (diagonal configurations $D_1$ and $D_2$) and $U_n \equiv U_{12} = U_{23} = U_{34} = U_{14}$ (nearest neighboring distributions $D_3 \ldots D_6$) separated by the energy gap $U \equiv U_n - U_d$ as discussed above. As to the states with two excess electrons per site, they are strongly excited and their mixing with the low-lying electronic configurations $D_1 \ldots D_6$ for which each site may contain no more than one excess electron gives rise to the HDVV exchange interaction between the metal ions. Below this HDVV exchange will be taken into account in the framework of the extended model (see next Section 3).

## 3. Extended Model: Exchange Interaction

As far as all ions in the considered system are magnetic, the basic model in general should be supplemented by the Heisenberg–Dirac–Van Vleck (HDVV) exchange, which will also be restricted to the pairs of nearest neighboring sites. In contrast to the double exchange determined by the electron transfer between the upper $\psi$-orbitals of the metal sites, the HDVV exchange also involves the $\varphi$-orbitals of the spin-cores and the electron transfer between these orbitals that contributes to the HDVV exchange coupling. Hereunder we will not discuss the microscopic mechanisms of HDVV exchange, assuming that all these mechanisms are fully incorporated in the exchange parameters $J_{i,k}$. The HDVV Hamiltonian acts within the full set of the states that can shortly be specified as

$\langle D_\lambda \big( \widetilde{S}(D_\lambda) \big) SM$ |. In this notations the set of the two intermediate spin values arising upon coupling of the four local spins are indicated as $\big( \widetilde{S}(D_\lambda) \big)$ (for example, this set can be $(S_{13}(D_\lambda), S_{24}(D_\lambda))$, $S$ is the quantum number of the total spin, and $M$ is the quantum number of the total spin projection. Note that the numbers of the electrons populating different sites and hence the local spins are defined by the electronic distribution. This is ensured by the symbol $D_\lambda$ which indicates that the set of spin functions belong to a certain distribution $D_\lambda$ of the mobile electrons ($\lambda = 1, 2..6$, Figure 1). Consequently, the intermediate spins in the four-spin coupling scheme are also defined by a certain distribution $D_\lambda$.

Since by definition the HDVV exchange acts within the system of localized spins it requires a following non-standard notation as applied to an MV system:

$$\hat{H}_{ex}(D_\lambda) = -2 \sum_{i<k, \sigma} J_{ik}(D_\lambda)(1 - \delta_{k, i+2}) \, \hat{\mathbf{S}}_i(D_\lambda) \hat{\mathbf{S}}_k(D_\lambda) \tag{2}$$

In the extended model the HDVV exchange Hamiltonian, Equation (2), should be added to the Hamiltonian, Equation (1), in order to obtain the full Hamiltonian of the tetramer. Such full Hamiltonian relates to the so-called *t-J* model. In Equation (2) the symbol $D_\lambda$ indicates that the exchange Hamiltonian $\hat{H}_{ex}(D_\lambda)$ acts within the space of spin-functions $\langle D_\lambda \, \widetilde{S}(D_\lambda), SM$| defined for a certain electronic configuration $D_\lambda$. This means that the matrix of $\hat{H}_{ex}$ has block-diagonal structure, $\langle D_\lambda \, \widetilde{S}'(D_\lambda), SM | \hat{H}_{ex}(D_\lambda) | D_\mu \, \widetilde{S}(D_\mu), SM \rangle \sim \delta_{\lambda\mu}$, where the Kronecker symbol $\delta_{\lambda\mu}$ ensures action of the exchange Hamiltonian within the set of spin states belonging to a definite distribution and excludes off-diagonal matrix elements. On the contrary, the double exchange Hamiltonian $\hat{H}_{DE}$ links states belonging to different distributions $D_\lambda$ so that $\langle D_\lambda \, \widetilde{S}'(D_\lambda), SM | \hat{H}_{DE} | D_\mu \, \widetilde{S}(D_\mu), SM \rangle \sim 1 - \delta_{\lambda\mu}$. The notation of spin-operators contains symbol $D_\lambda$ of configuration in addition to the running symbol $i$ numerating the sites that defines the value of $s_i$. For example, for the distribution $D_1$ (Figure 2) $s_1 = s_3 = s_0 + 1/2$, while the two remaining sites have spins $s_0$. Each distribution $D_\lambda$ generates a specific network of the exchange interactions whose parameters are reduced to the three independent quantities, $J\big(d^{n+1} - d^n\big) \equiv J$, $J\big(d^{n+1} - d^{n+1}\big) \equiv J_1$ and $J(d^n - d^n) \equiv J_2$ as illustrated in Figure 2a,b for particular cases of distributions $D_3$ and $D_1$.

## 4. Combined Effect of the Double Exchange and Coulomb Repulsion

We will illustrate the main features of the double exchange in a bi-electronic system considering a simple case of tetrameric unit $d^2 - d^2 - d^1 - d^1$- type in which the paramagnetic spin-core contains the only unpaired electron ($s(d^1) = s_0 = 1/2$, $s\big(d^{n+1}\big) = 1$). The total spin of such tetramer consisting of two spins 1/2 and two spins 1, takes the values: $S = 3, 2, 1, 0$. The calculation of the energy spectrum can be performed with the aid of angular momentum approach (see review articles) [31,32] and MVPACK program based on this approach [33]. In order to elucidate the effect of the double exchange we present in Figure 3 the energy spectrum calculated within the basic model taking into account only the double exchange and the Coulomb repulsion (the HDVV exchange will be included in consideration later on).

The main result that follows from Figure 3 is that the ground state of the MV tetramer proves to be diamagnetic independently of the relative strength of the Coulomb repulsion and the double exchange. At first glance this result seems to be unexpected and contradicts the basic paradigm of the double exchange. Indeed, in the ground state of the system the spin of mobile electron is aligned parallel to the spin core due to ferromagnetic intraatomic exchange coupling. While moving the mobile electron polarizes the spin cores aligning them in a parallel fashion thus giving rise to ferromagnetic effect in MV systems. Such effect is quite general and, particularly, it occurs in MV dimers [3] and square-planar MV tetramers [34,35] comprising one excess electron or hole in which the double exchange always leads to the ferromagnetic ground state (a special effect of frustration in MV systems discovered in Ref. [36] is out of the scope of this topic). To elucidate the physical origin of the antiferromagnetic effect of the double exchange in the $d^2 - d^2 - d^1 - d^1$ tetramer let us imagine that this MV system is divided

into two interacting parts, $d^1 - d^1 - d^1 - d^1$ core (with uncoupled spins) and a bi-electronic tetramer $d^1 - d^1 - d^0 - d^0$ in which two electrons are delocalized among four diamagnetic centers.

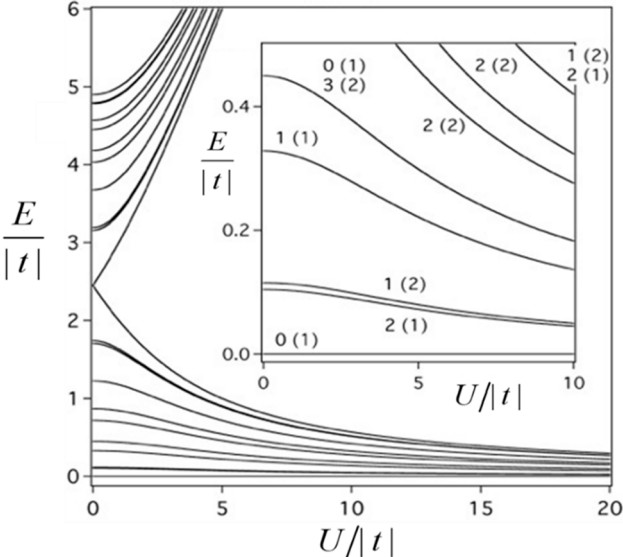

**Figure 3.** Combined effect of double exchange and Coulomb interaction on the energy spectrum of square-planar $d^2 - d^2 - d^1 - d^1$– tetramer. The low-lying part of the energy spectrum with labelling of the energy levels is shown as insert. The energy levels are labelled as $S$ $(f)$, where $S$ is the total spin of the tetramer and $f$ is multiplicity of the repeated levels with the same $S$. The energy of the ground state is regarded as a reference energy.

The energy pattern of the delocalized electronic pair in a square [29] shows that the ground state is the spin-singlet with the energy $\left(U - \sqrt{U^2 + 16t^2}\right)/2$, while the next level involves the two spin-triplets with the energy $\left(U - \sqrt{U^2 + 32t^2}\right)/2$. Within the imaginative classical picture, the two excess electrons in the ground state always keep their spins antiparallel in course of the electron delocalization. This is also valid when the two electrons are delocalized over the network of spin cores and hence the electron which keeps its spin "up" tends to polarize the core spins also "up", while the second electron with spin "down" produces opposite effect. This is schematically shown in Figure 4a from which one can see that the delocalization of the two spins of opposite directions results in the full compensation of the magnetic moments and consequently to the antiferromagnetic ground state of the square planar MV cluster with two excess electrons. On the contrary, when the delocalized unit is in the spin-triplet excited state the two electrons have parallel spin and the system entire becomes ferromagnetic as shown in Figure 4b. This qualitative picture of manifestation of the double exchange in multielectron MV systems is rather general, and in particular, is applicable to the systems $d_1^{n+1} - d_2^{n+1} - d_3^n - d_4^n$ with arbitrary spin cores.

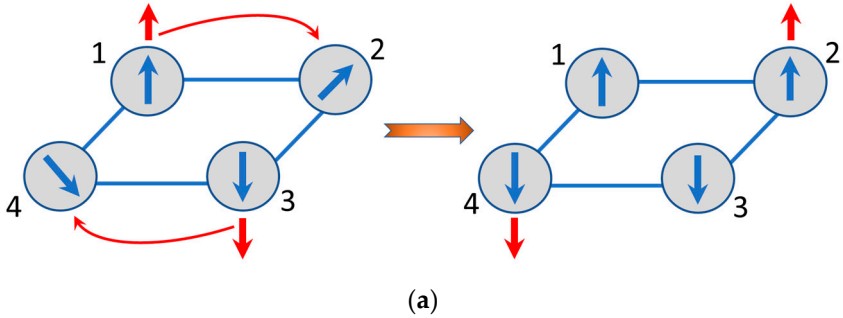

(**a**)

**Figure 4.** *Cont.*

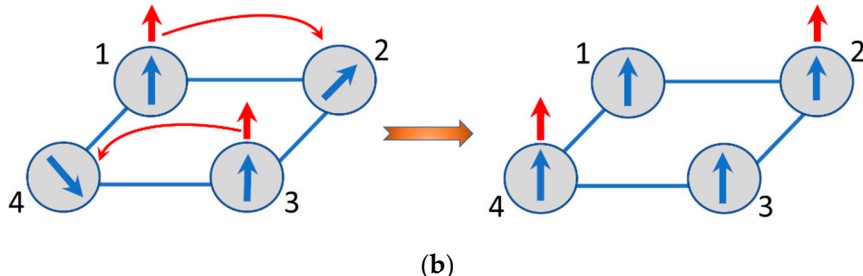

**(b)**

**Figure 4.** Illustration for the spin polarization effects in the tetrameric square-planar systems with two mobile electrons: antiferromagnetic spin alignment of delocalized spins (**a**); ferromagnetic spin alignment of delocalized spins (**b**).

## 5. Double Exchange in Regime of Strong Coulomb Repulsion

We will focus on the topical case of strong Coulomb repulsion which is relevant to potential application of the bielectronic MV square as a molecular cell for QCA. In this case the low-lying group of levels corresponding to the antipodal electronic distributions and the excited levels arising from the neighboring configurations are well separated from each other by the Coulomb gap $U$ (Figure 1) which considerably exceeds the transfer parameter $t$ and all exchange parameters $J$, $J_1$ and $J_2$ (Figure 2). In this case, which we will refer to as a strong $U$ limit, one can apply the perturbation theory with the Coulomb term $\hat{H}_0$ playing the role of zero-order Hamiltonian and the operator $\hat{V}$ acting as perturbation. In the strong $U$ limit the non-vanishing effect of the double exchange appears within the second order of perturbation theory, while the first order terms vanish. This is visualized in Figure 5 from which one can see that at the first step the initial antipodal configuration $D_1$ passes into neighboring position $D_6$ via one-electron transfer $1 \rightarrow 2$ and then the jump $3 \rightarrow 4$ transforms $D_6$ into the final antipodal configuration $D_2$. One can see that the first order transfer does not operate within the space of only distant configurations, while the second order process does. That is why the effective bi-electronic transfer parameters that appear in the second order perturbation calculations is expressed as $\tau = t^2/U$. The second order double exchange separates the energy levels according to the full spin of the system giving rise to the two states with $S = 3$, six states with $S = 2$, eight states with $S = 1$ and four states with $S = 0$. The energies of these states are listed in the Table 1.

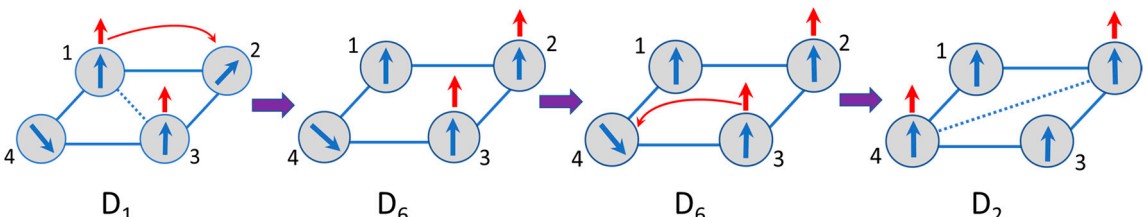

**Figure 5.** Visual representation of the sequential $1 \rightarrow 2$, $3 \rightarrow 4$ double electron transfer $D_1 \rightarrow D_2$ which acts within the basis spin-functions belonging to the two antipodal localization of the electronic pair in MV tetramer of $d_1^{n+1} - d_2^{n+1} - d_3^n - d_4^n$ type. Spin polarization effect is also schematically shown.

**Table 1.** Spin levels $E(S)/\tau$ belonging to the antipodal charge configuration in the strong $U$ limit. The numbers of the levels having the same spin and energy are indicated in parenthesis.

| $S$ | $E(S)/\tau$ |
|:---:|:---:|
| $S = 3$ | $-4$ (2) |
| $S = 2$ | $-5/4(2); -3(2); -11/2(1); -3/2$ (1) |
| $S = 1$ | $-3/2$ (2); $-1/2$ (1); $-9/2$ (1); $\sqrt{6}$ (2); $-\sqrt{6}$ (2) |
| $S = 0$ | $-4$ (1); $0(2); -6(1)$ |

In compliance with the previous result the second order double exchange stabilizes the antiferromagnetic ground state. One can see that the spin states are degenerate, for example, the $S = 3$ level is repeated twice accordingly to the presence of the two antipodal localizations. As has been reported for the well-known examples of symmetric MV clusters, such multiple degeneracy can arise also from the orbital degeneracy of the terms that can be established by the use of the group-theoretical assignation. As a simple example one can consider in more detail the doubled state with maximal spin $S = 3$ which definitely arise from the coupling of the $S = 2$ state of the one-electron tetramer and spin-triplet state of the pair. In the strong $U$-limit the last was attributed to the orbital doublet $^3E$ in $\mathbf{D}_{4h}$ group (which is the excited level) while the $S = 2$ ($S_{13} = 1, S_{24} = 1$) can be designated as $^5B_{1g}$ term. Then by coupling the states of the localized and delocalized units one can conclude that the term of the $d^2 - d^2 - d^1 - d^1$ tetramer with the maximal spin $S = 3$ is $^7E$. One can see from the Table 1 that the degeneracy of two $S = 3$ states is related to the orbital degeneracy which means that this is an "exact" degeneracy originating from the point symmetry of the system. A comprehensive discussion of the degeneracies in spin systems and their physical consequences can be found in review articles [31,32]. In particular, one can observe the so-called "accidental degeneracy" interrelated with the unitary symmetry that in general are high than the point one. Regarding the action of the double exchange, one can conclude that two electrons in $^3E$ term produce ferromagnetic double exchange in the system (as schematically shown in Figure 4b).

## 6. Beyond Basic Model: Concomitant Effect of the HDVV Exchange

While the conceptual features of the double exchange in MV system under consideration can be described within the basic model, the HDVV exchange plays an important role in the case of strong $U$ because the exchange parameters can be comparable with the residual (second order) double exchange parameter $\tau = t^2/U$. For this reason, now we will briefly examine a combined effect of the second-order double exchange and the HDVV exchange in the strong $U$ limit. This can be called "$\tau$—$J$ model". Considering qualitatively the role of the HDVV exchange, it is worth to underline that the double exchange solely aligns the non-interacting spins in the core network. Since they are free, each spin core is able to freely adapt its direction along the mobile spin giving rise to the spin alignment in the whole system. In contrast, the HDVV exchange itself aligns spins in each localized configuration, and so the mobile electron polarizes already partially ordered (but not free) subsystem of spin cores. This predetermines special spin dependence of the energy pattern of MV systems when the HDVV exchange is operative, as well as its dependence on the parameters $J$ and $\tau$.

In compliance with numerous data on the magnetic properties of transition metal complexes we assume that the HDVV exchange is antiferromagnetic ($J < 0$). According to the general definition in Equation (2) the exchange Hamiltonian acting within the distant diagonal distributions $D_1$ and $D_2$ can be written as:

$$\hat{H}_{ex}(D_\lambda) = -2J\left(\hat{S}_1(D_\lambda)\hat{S}_2(D_\lambda) + \hat{S}_2(D_\lambda)\hat{S}_3(D_\lambda) + \hat{S}_3(D_\lambda)\hat{S}_4(D_\lambda) + \hat{S}_4(D_\lambda)\hat{S}_1(D_\lambda)\right) \qquad (3)$$

where $\lambda = 1, 2$ and the spin values are determined for each of the two distributions as described above. The network of exchange parameters in the two distant configurations is reduced to the only parameter $J(D_1) = J(D_2) = J\left(d^{n+1} - d^n\right) \equiv J$ as illustrated in Figure 2b, while the spin values are determined for each distribution: $S_1 = S_3 = s_0 + 1/2$, $S_2 = S_4 = s_0$ for configuration $D_1$ and $S_1 = S_3 = s_0$, $S_2 = S_4 = s_0 + 1/2$ for $D_2$. The energy levels are expressed in terms of the full spin $S$ and the two intermediate spins $S_{13}$, $S_{24}$ which are peculiar for each distribution:

$$E_{ex}(S_{13}, S_{24}, S) = -J[S(S + 1) - S_{13}(S_{13} + 1) - S_{24}(S_{24} + 1)]. \qquad (4)$$

The energies of the system as functions of $J$ and $\tau$ are shown in correlation diagram (Figure 6). It is seen that depending on the relative strength of the second-order double exchange and the HDVV

exchange the ground state can be either diamagnetic, $S = 0$ (for $\tau/|J| > 4/5$) or magnetic with $S = 1$ (when $\tau/|J| < 4/5$), while the remaining allowed spin values correspond to the excited states.

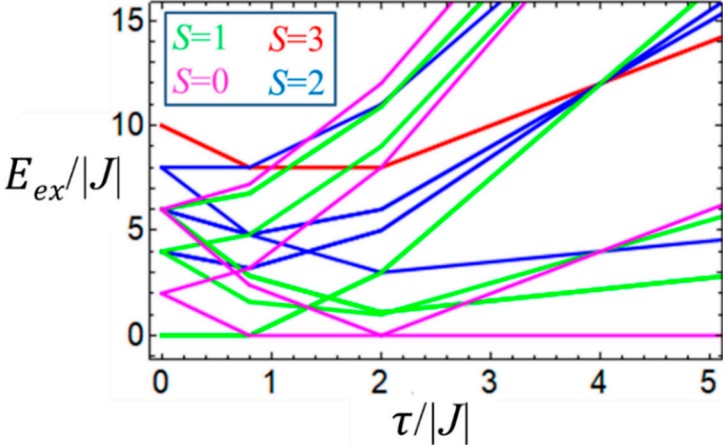

**Figure 6.** Correlation diagram $E_{ex}/|J|$ *vs* $\tau/|J|$ for the MV square-planar $d^2 - d^2 - d^1 - d^1$ -tetramer calculated within the strong $U$-limit. Coloring of spin states is shown in the insert.

Stabilization of diamagnetic ground state at strong enough second-order double exchange makes the properties of the square planar system somewhat similar to the properties of the two-electron reduced polyoxometalates with Keggin structure in which the combined action of electron delocalization and the intersite Coulomb repulsion can result in the diamagnetic ground state [6]. On the other hand, the square planar system with two mobile electrons exhibits magnetic behavior which is quite different from the magnetic behavior of MV dimers or magnetic square planar MV tetramers with the only mobile electron in which the double exchange interaction is known to stabilize the ferromagnetic ground state.

## 7. Conclusions

We have studied the energy pattern of a square-planer tetrameric MV system of $d^{n+1} - d^{n+1} - d^n - d^n$ type in which both double exchange and HDVV exchange are operative and the Coulomb repulsion between the two electrons is taken into account. The detailed results are given for the model system $d^2 - d^2 - d^1 - d^1$ in which the two electrons are shared among for spin-1/2 cores. It is demonstrated that at zero HDVV exchange the ground state of this system is diamagnetic irrespectively of the interrelation between the second order double exchange and Coulomb energy. A special attention is paid to the case of strong Coulomb repulsion which is important for the potential application of this system as molecular cell of QCA device.

Summing up this consideration we can go back to the question in title: can the double exchange cause antiferromagnetic spin alignment? The double exchange as a basic mechanism of spin alignment arising from spin polarization is undoubtedly ferromagnetic as prescribed by the Hund rule. That is why in MV systems having the only mobile electron the double exchange produces ferromagnetic effect (excluding special cases of frustration). In contrast, the action of this mechanism in systems comprising two excess electrons can result in the antiferromagnetic effect due to opposite spin directions in the subsystem of mobile electrons which can lead due to stratification of the full MV system into subsystems with opposite spin directions as was demonstrated through a case study.

**Author Contributions:** Conceptualization, A.P., B.T., S.Z.; methodology, A.P., S.Z.; software, D.K., J.M.C.-J.; validation, D.K., J.M.C.-J.; writing—original draft preparation, A.P., B.T., S.A.; writing—review and editing, A.P., B.T., S.A.; visualization, E.G.; supervision, A.P., S.A., B.T.; project administration, E.G.; funding acquisition, A.P., B.T. All authors have read and agreed to the published version of the manuscript.

**Funding:** This research was funded by Russian Science Foundation grant number [20-13-00374]. The APC was funded by MDPI-Magnetochemistry.

**Acknowledgments:** Support from Russian Science Foundation (project No. 20-13-00374) is acknowledged.

**Conflicts of Interest:** The authors declare no conflict interest.

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
