# Peer review of "Can the Double Exchange Cause Antiferromagnetic Spin Alignment?"

_magnetochemistry, doi:10.3390/magnetochemistry6030036_

Round 1
Reviewer 1 Report
In this manuscript, the authors study the effect of the double exchange in a square-planar mixed-valence (MV) tetramers of the general type d^(n+1) − d^(n+1) − d^n − d^n, with two excess electrons delocalized over four spin cores. Using the case of n=1 as a model case, they study, in detail the tetramer d^2 − d^2 − d^1 − d^1. They show that while the double exchange interaction gives a ferromagnetic configuration in MV systems with only one mobile electron, systems with 2 excess electrons can result in an antiferromagnetic configuration.
The topic of the manuscript is within the scope of the journal, and is certainly of interest to its audience. The manuscript is nicely written, with a smooth and comprehensible flow. All figures are clear and of adequate resolution. I recommend the acceptance of the manuscript in its present form.
Author Response
We are grateful to the Reviewer for his positive opinion about our article.
Reviewer 2 Report
This work describes the effect of second-order double exchange and Heisenberg-Dirac-Van Vleck exchange on a model square-planer tetrameric multi-valent system. It was shown qualitatively and quantitatively that the ground state (which can be a singlet or a triplet) is determined by the relative magnitudes of the two exchange terms. The work seems interesting and the paper is well written. I have minor suggestions for improving the paper:
In the abstract,
the system entire becomes antiferromagnetic. --> the system entirely becomes antiferromagnetic.
In the introduction, in the following sentence, “…hexanuclear octahedral clusters [Fe6(μ3-X)8(Pet3)6]+(X=S, Se)…”, the full form of Pet needs should be given.
In Section 3. Extended model: exchange interaction, what is M in ⟨??(?̃(?))?M |?
In Section 3, has bock-diagonal structure, --> has block-diagonal structure,
In Section 3, in Figure 3 caption, (a) is mentioned but (b) is not mentioned.
After the above suggestions are implemented, the paper can be accepted.
Author Response
Reviewer_2
This work describes the effect of second-order double exchange and Heisenberg-Dirac-Van Vleck exchange on a model square-planer tetrameric multi-valent system. It was shown qualitatively and quantitatively that the ground state (which can be a singlet or a triplet) is determined by the relative magnitudes of the two exchange terms. The work seems interesting and the paper is well written. I have minor suggestions for improving the paper:
In the abstract,
the system entire becomes antiferromagnetic. --> the system entirely becomes antiferromagnetic.
This is corrected.
In the introduction, in the following sentence, “…hexanuclear octahedral clusters [Fe6(μ3-X)8(Pet3)6]+(X=S, Se)…”, the full form of Pet needs should be given.
It was a misprint in the text (Pet instead of PEt). Now it is improved and the full form of Et is given.
In Section 3. Extended model: exchange interaction, what is M in ⟨??(?̃(?))?M |?
Now M is explained.
In Section 3, has bock-diagonal structure, --> has block-diagonal structure,
It is corrected.
In Section 3, in Figure 3 caption, (a) is mentioned but (b) is not mentioned.
Now (b) is mentioned as well.
After the above suggestions are implemented, the paper can be accepted.
Reviewer 3 Report
Here the authors try to prove double exchange cause antiferromagnetic order in a square-planar mixed system using the model Hamiltonian method. I think this manuscript is not presented well. And, I have some comments for the authors to consider.
1, Could the authors show what the model they use in Fig.1. Why the energies of D1 and D2 configuration are lower than energies of D3, D4, D5, and D6 by U
2, The point group of square-planar is D4h, I think D1 and D2 should be the same configurations, why the authors think they are different. The same question for D3,D4,D5 and D6 configurations., they should be the same.
3, In Fig.2, every site should have at least two orbitals. Why the authors use a single orbital Hubbard model rather than a multi-orbit Hubbard model.
4, What is the relationship between eq.1 and eq.2? The authors use HDVV exchange model to describe tetramer. Could the authors put more words about why HDVV work here? And what is the difference between HDVV and RKKY exchange?
Author Response
Reviewer_3
Here the authors try to prove double exchange cause antiferromagnetic order in a square-planar mixed system using the model Hamiltonian method. I think this manuscript is not presented well.
Reply: We have improved presentation of the model in order to make it clearer.
And, I have some comments for the authors to consider.
- Could the authors show what the model they use in Fig.1. Why the energies of D1 and D2 configuration are lower than energies of D3, D4, D5, and D6 by U?
Reply: The two excess electrons can be distributed in six different ways over four centers (the configurations with two excess electrons occupying the same site are much higher in energy and they are excluded from the consideration). Due to the fact that there are two different intersite distances in the system of D4h symmetry (diagonal of the square and the side of the square), there are also two different Coulomb energies of electronic repulsion, namely lower energy for the distant electrons when they occupy vertices of the square situated along the diagonal and higher energy when the two electrons occupy nearest neighboring positions (side of the square). The difference between these two energies is denoted by U. For this reason, the energies of D1 and D2 configuration (diagonal dispositions of the two electrons) are lower than energies of D3, D4, D5, and D6 (side dispositions) by U. In the revised article this point has been carefully explained (see pages 3, 4).
- The point group of square-planar is D4h, I think D1 and D2 should be the same configurations, why the authors think they are different. The same question for D3, D4, D5 and D6 configurations., they should be the same.
Reply: As has been mentioned above (reply to comment 1) and also in the revised article, the two “diagonal“ configurations D1 and D2 are energetically equivalent. The same is true for four “nearest neighboring” configurations D3, D4, D5 and D6 (these four configurations have the same energies). At the same time the Coulomb energies of diagonal and nearest neighboring configurations are different (answer to comment 1 and also explanation in the revised article). This difference in Coulomb energies does not contradict to the D4h symmetry of the complex because the two ground diagonal configurations are transformed into each other under the action of the operation of rotation around C4 axis within the D4h point group, as well as the four excited configurations. At the same time the excited configurations cannot be obtained from the ground ones by the D4h group operations thus showing that these two kinds of configurations are physically different. This has been additionally discussed in the revised article (see comment at the beginning of page 4).
- In Fig.2, every site should have at least two orbitals. Why the authors use a single orbital Hubbard model rather than a multi-orbit Hubbard model.
Reply: This concern of the Reviewer is caused by the fact that in initial version of the article the model has not been clearly described. In fact, we do not use the Hubbard Hamiltonian although the Hamiltonian in Eq. (1) looks like Hubbard Hamiltonian. The genuine Hubbard Hamiltonian acts within the space comprising configurations with double occupation of the sites and the Coulomb interaction term in Hubbard Hamiltonian describes the on-site Coulomb repulsion. In contrast, the Coulomb term in our Hamiltonian, Eq. (1), describes repulsion between the excess electrons occupying different metal sites (intersite Coulomb repulsion). The Hamiltonian, Eq. (1), can be regarded as a double exchange Hamiltonian in clusters in which different distributions of the electronic pair possess different intersite Coulomb energies. When the HDVV exchange is also included, our Hamiltonian corresponds to the t-J model. In response to the Reviewer comment we carefully explained the difference between these two Hamiltonians in the revised article (page 5 after Eq. (1)). In order to clarify why only one (upper) orbital in each metal site is involved in the electron transfer process we have added the orbital scheme (new Fig. 2c) showing such electron transfer and explained the background of the model of double exchange conventionally used for MV clusters (see explanations at the end of page 4 and also in the beginning of Section 3).
- What is the relationship between eq.1 and eq.2? The authors use HDVV exchange model to describe tetramer. Could the authors put more words about why HDVV work here? And what is the difference between HDVV and RKKY exchange?
Reply: In reply to the Reviewer’s comment we improved the description in the initial version of the article. In fact, the HDVV Hamiltonian, Eq. (2), should be added to the double exchange Hamiltonian, Eq. (1), to obtain the full Hamiltonian of the system. This was clearly indicated in the revised manuscript (comment after Eq. (2) in page 6). The RKKY exchange is somewhat similar to the double exchange because it can be regarded as a limiting case of double exchange when the electron transfer is much stronger than the intraatomic exchange. This is not evidently the case in considered class of clusters when the energy gap between the Hund and non-Hund states of the constituent ions considerably exceeds the electron transfer. We have commented this in the revised article (page 4). As to the HDVV exchange, it acts within each localized electronic configuration and forms the diagonal (with respect to the electronic configurations) block of the full Hamiltonian matrix, while the double exchange is off-diagonal with respect to such configurations because it is associated with the electron transfer changing the localization positions of the excess electrons.
We are thankful to the reviewer for his valuable comments that allowed us to improve the manuscript.
Round 2
Reviewer 3 Report
The revised version is much better. I recommend it for publication.